# First-line durvalumab therapy alone or in combination with tremelimumab for metastatic head and neck squamous cell carcinoma: A cost-effectiveness analysis

Huijuan Li[1,2,3☉], Xueyan Liang[2☉], Shiran Qin[3], Xiaoyu Chen[2,3], Liju Huang[2*], Yan Li[ID][3*]

**1** Drug Clinical Trial Institution, Guangxi Academy of Medical Sciences and the People's Hospital of Guangxi Zhuang Autonomous Region, Nanning, Guangxi, People's Republic of China, **2** Phase I Clinical Trial Laboratory, Guangxi Academy of Medical Sciences and the People's Hospital of Guangxi Zhuang Autonomous Region, Nanning, Guangxi, People's Republic of China, **3** Department of Pharmacy, Guangxi Academy of Medical Sciences and the People's Hospital of Guangxi Zhuang Autonomous Region, Nanning, Guangxi, People's Republic of China

☉ Huijuan Li and Xueyan Liang contributed equally to this study.
\* yli@gxams.org.cn (YL); 395154496@qq.com (LH)

## Abstract

### Objective

In patients with recurrent or metastatic head and neck squamous cell carcinoma (R/M HNSCC), the KESTREL study evaluated durvalumab with or without tremelimumab, as well as the EXTREME regimen (platinum, 5-fluorouracil, and cetuximab). In the first-line treatment of R/M HNSCC, no data are available regarding the cost-effectiveness of these immunotherapeutic agents. Based on third-party payers' perspectives in the United States, we compared durvalumab and tremelimumab in this setting.

### Methods

The projected costs and outcomes over a 12-year period were calculated using a three-state partitioned survival model with an annual discount of 3%. Long-term extrapolation of KESTREL was used to model progression-free and overall survival. Life-years, quality-adjusted life-years (QALYs), incremental cost-effectiveness ratios (ICERs), incremental net monetary benefits (INMBs), and incremental net health benefits (INHBs) were calculated. The uncertainty and stability of the model were examined using one-way and probabilistic sensitivity analyses.

### Results

In the overall population, EXTREME was dominant compared to durvalumab plus tremelimumab (Durva-Treme). Compared to durvalumab monotherapy (Durva-mono), EXTREME resulted in an increase of 0.037 life-years and 0.012 QALYs, as well as

**Data availability statement:** The clinical and safety data were derived from a recently randomized clinical trial (KESTREL). Utilities were collected from the published literature included in PubMed and Web of science. Costs were obtained from CMS (https://www.cms.gov/medicare/medicare-part-b-drug-average-sales-price/2023-asp-drug-pricing-files) and Lexicomp Online (https://www.wolterskluwer.com/en/solutions/lexicomp/lexicomp).

**Funding:** This study was supported by the Drug Safety Research Project of Guangxi Zhuang Autonomous Region (Guiyaojiankezhishu[2023]017).

**Competing interests:** The authors have declared that no competing interests exist.

an increase in cost of $85,420 per patient. The corresponding ICER was $7,310,878/QALY, and the INHB and INMB values were negative at a willingness to pay threshold of $150,000/QALY. Furthermore, Durva-mono dominated EXTREME or Durva-Treme in patients with high PD-L1 expression. When compared with EXTREME, Durva-Treme had an ICER of $560,406/QALY. Durva-Treme was unlikely to be cost-effective. The HRs for OS, PFS, and drug costs (cetuximab, tremelimumab, and durvalumab) were most sensitive to the model. The results of the sensitivity analyses were similar, suggesting the robustness of the findings.

## Conclusions

Durva-mono is a cost-effective treatment strategy for patients with R/M HNSCC, particularly in patients with high PD-L1 expression, compared with EXTREME or Durva-Treme. Cetuximab cost reduction may benefit the EXTREME regimen economically. Nevertheless, Durva-Treme provided fewer survival gains at substantial additional costs; therefore, it is unlikely to be a cost-effective option.

## Introduction

The organs involved in head and neck cancer include the oral cavity, nasal cavity, pharynx, larynx, salivary glands, and paranasal sinuses [1]. Head and neck cancers are predominantly squamous cell carcinoma [2]. Most patients diagnosed with head and neck squamous cell carcinoma (HNSCC) have locally advanced cancer, and more than half experience recurrence or metastasis [3]. Multiple conventional treatment regimens do not improve the overall survival (OS) of patients with platinum-based chemotherapy-recurrent or metastatic HNSCC (R/M HNSCC). Consequently, new treatment options are urgently needed because of the lack of increased OS and the high incidence of adverse events [4].

A landmark success in the treatment of head and neck cancers has been the use of immunotherapy in the form of checkpoint inhibitors. For the treatment of R/M HNSCC, the European Medicines Agency and Food and Drug Administration have approved the anti-PD-1 antibodies pembrolizumab and nivolumab as single agents [5–8]. Multiple mechanisms may contribute to HNSCC evasion of antitumor immunity, including the frequent upregulation of immune checkpoint molecules such as PD-1, programmed death-ligand 1 (PD-L1), and cytotoxic T-lymphocyte-associated antigen 4 (CTLA-4) [9]. In addition to being a monotherapy for HNSCC, durvalumab is a safe and effective anti-PD-L1 antibody [10,11]. Additionally, tremelimumab is a safe anti-CTLA-4 antibody [12]. Durvalumab and tremelimumab were recently studied as first-line treatments for R/M HNSCC in the KESTREL phase III trial [13]. Durvalumab monotherapy (Durva-mono) and durvalumab-tremelimumab combination (Durva-Treme) therapy resulted in longer response durations (49.3% and 48.1%, respectively) than the EXTREME regimen (platinum plus cetuximab plus 5-fluorouracil, 9.8%). Considering OS, in patients with high PD-L1 expression (immune cell ≥25% or

tumor cell ≥50%), Durva-mono and Durva-Treme were not superior to EXTREME; however, Durva-mono and EXTREME were comparable. Therefore, Durva-mono (8.9%) and Durva-Treme (19.1%) regimens were associated with a lower incidence of grade 3 treatment-related adverse events than was the EXTREME regimen (53.1%).

Notably, immune therapy is often associated with higher medical costs than traditional chemotherapy, despite its proven clinical efficacy. Therefore, economically evaluating this novel drug is imperative. Durvalumab with or without tremelimumab has not yet been evaluated as a cost-effective treatment for HNSCC compared to the EXTREME regimen, despite the fact that other immune checkpoint inhibitors, such as nivolumab and pembrolizumab have been studied [1,14]. Therefore, based on the results of the KESTREL trial, we aimed to determine whether these regimens were cost-effective for R/M HNSCC treatment in the United States, either in the total population or in subgroups with high PD-L1 expression (TC ≥ 50% or IC ≥ 25%).

## Materials and methods

### Patients and intervention

As no human participants were involved in this economic evaluation study based on the KESTREL trial [13], the institutional review board exempted the study from ethical approval. This study was conducted according to the Consolidated Health Economic Evaluation Reporting Standards 2022 (CHEERS 2022) reporting [15]. This cohort was the same as that of the KESTREL trial, which included a target patient population. Patients were aged ≥ 18 years and had histologically or cytologically confirmed R/M HNSCC that was not amenable for treatment by local radiation or surgery. PD-L1 expression was evaluated using the VENTANA PD-L1 (SP263) assay-based immunohistochemistry. PD-L1 high was defined as either ≥50% of TCs or ≥25% of ICs staining positive for PD-L1 at any intensity if >1% of the tumor area contained ICs or ≥50% of TCs or 100% of ICs staining for PD-L1 at any intensity if 1% of the tumor area contained ICs. PD-L1 low was defined as a condition that did not meet any of the characteristics of PD-L1 high. The VENTANA PD-L1 (SP263) assay has not been approved for any other detection method or instrument, and only rabbit monoclonal negative control Ig can be used.

Three groups were included in this study: durvalumab monotherapy (Durva-mono, 1500 mg), durvalumab plus tremelimumab (Durva-Treme, 75 mg), and the EXTREME regimen. In the EXTREME regimen, cisplatin was administered at 100 mg/m$^2$ of body surface area or carboplatin was administered at 5 mg/mL/min, and 5-fluorouracil was administered at 1000 mg/m$^2$/d. In addition, cetuximab was administered at 400 mg/m$^2$ on day 1 and 250 mg/m$^2$ weekly thereafter.

### Partitioned survival model

The three competing regimens were compared among patients with R/M HNSCC using a partitioned survival model. Three health states were modeled: progression-free survival (PFS), progressed disease (PD), and death. The PFS state was the initial state of health for all patients, and during each cycle, patients could maintain their assigned health state or redistribute it to another health state (Fig 1) [16]. At each time point, the area under the curve (AUC) of the PFS was used to estimate the proportion of patients in the PFS state, and 1 minus the OS curve was used to estimate the proportion of patients who died. The PD state was measured using the AUC between the PFS and OS curves. A one-week cycle length was chosen to facilitate parameter calculations. To ensure that patients with R/M HNSCC reached the terminal stage, a 12-year time horizon was devised (≥ 98% of the patients died by this time).

### Clinical data inputs

Clinical efficacy and safety data were collected based on the KESTREL trial [13]. Due to the lack of individual patient data (IPD), PFS and OS data points were extracted from Kaplan-Meier(K-M) survival curves using GetData Graph Digitizer (version 2.26) [17]. To extrapolate the survival curves beyond the follow-up duration of the clinical trials, various parametric distributions were fitted, including Gamma, Exponential, Weibull, Lognormal, Log-logistic, Gompertz, and Generalized

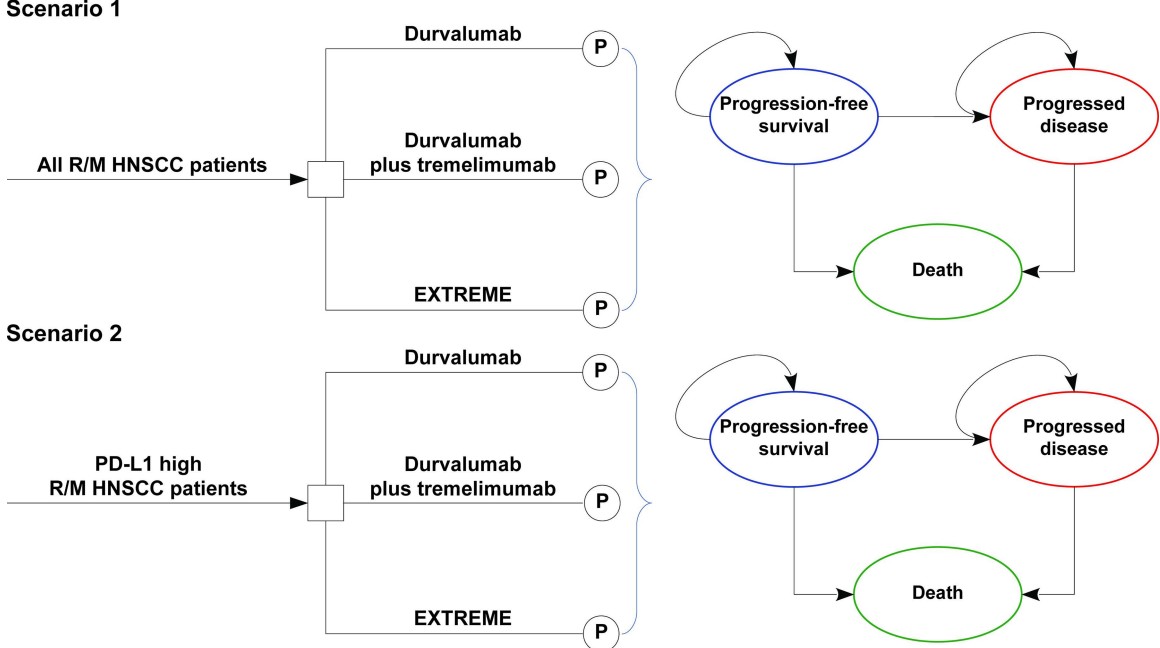

**Fig 1. Partitioned survival model with three discrete health states.**

gamma. The distribution with the best fit was identified using the Bayesian information criterion (BIC), Akaike information criterion (AIC), and graphical validation (S1 Table). The AIC and BIC were calculated based on survival analyses using R version 4.0.5. A comparison of the model-fitted versus the original K-M curves is shown in S1 Fig. The key clinical inputs are listed in Table 1 [2,13,18–26].

## Costs

Drugs, routine follow-up, laboratory tests, best supportive care, management of adverse events (AEs), and end-of-life terminal care were considered in the analysis. The unit price of tremelimumab was calculated using Lexicomp Online [19]. The prices of other drug unit were extracted from the Center for Medicare and Medicaid Services (CMS) [18]. The cost of terminal care per patient is $11126 [20]. Based on Tom's Inflation Calculator [27], all costs were inflated to 2022 US dollars using Medical-Care Inflation data, and the values are shown in Table 1. The average body surface area (BSA, 1.86 m²) and body weight (70 kg) were used to estimate the chemotherapy dosage [2].

## Utilities

An anchored utility value was assigned to each health state in this partitioned survival model, with 0 indicating death and 1 indicating perfect health. In the absence of data from the KESTREL trial, we derived the utility values from published sources. A PD health utility value of 0.749 and PFS health utility value of 0.805 were associated with HNSCC [24]. Furthermore, grade ≥3 AEs disutility values were considered in accordance with relevant literature (S2 Table) [22,23,25,26].

## Base-case analysis

The treatment groups were compared using various measures, including overall costs, life years, quality-adjusted life years (QALYs), incremental cost-effectiveness ratios (ICERs), incremental net monetary benefit (INMB), and incremental

**Table 1. Key model inputs.**

| Parameter | Value (95% CI) | Distribution | Source |
|---|---|---|---|
| OS survival model of durvalumab plus tremelimumab[a] | μ = 3.7767<br>σ = 1.2548 | Lognormal | 13 |
| PFS survival model of durvalumab plus tremelimumab[a] | γ = 1.4830<br>λ = 0.0753 | Log-logistic | 13 |
| OS survival model of durvalumab[a] | μ = 3.7974<br>σ = 1.2199 | Lognormal | 13 |
| PFS survival model of durvalumab[a] | γ = 1.5989<br>λ = 0.0796 | Log-logistic/ Lognormal | 13 |
| OS survival model of EXTREME[a] | μ = 3.8660<br>σ = 1.1712 | Lognormal | 13 |
| PFS survival model of EXTREME[a] | γ = 2.1616<br>λ = 0.0464 | Log-logistic | 13 |
| **Drug costs per 1 mg** | | | |
| Durvalumab | 7.89 (6.31 to 9.47) | Gamma | 18 |
| Tremelimumab | 3120 (2496–3744) | Gamma | 19 |
| Cisplatin | 0.17 (0.14 to 0.20) | Gamma | 18 |
| Carboplatin | 0.05 (0.04 to 0.06) | Gamma | 18 |
| 5-fluorouracil | 0.005 (0.004 to 0.006) | Gamma | 18 |
| Cetuximab | 7.04 (5.63 to 8.44) | Gamma | 18 |
| Cost of terminal care per patient[b] | 11126 (8901–13351) | Gamma | 20 |
| Best supportive care cost per cycle | 4645 (3716–5574) | Gamma | 2 |
| **Drug administration cost** | | | |
| First hour | 150.34 (123.55 to 198.06) | Gamma | 21 |
| Additional hour | 32.23 (27.29 to 41.02) | Gamma | 21 |
| **Cost of managing AEs (grade ≥ 3)[c]** | | | |
| Durvalumab plus tremelimumab | 14425 (11540–17310) | Gamma | 22, 23 |
| Durvalumab | 9751 (7801–11701) | Gamma | 22, 23 |
| EXTREME | 39108 (31286–46930) | Gamma | 22, 23 |
| Immunohistochemical test (per time) | 76.45 (35.57 to 146.77) | | 21 |
| Follow-up cost per cycle | 380 (304–456) | | 2 |
| **Health utilities** | | | |
| **Disease status utility per y** | | | |
| Utility PFS | 0.805 (0.644 to 0.966) | Beta | 24 |
| Utility PD | 0.749 (0.599 to 0.899) | Beta | 24 |
| Death | 0 | NA | |
| **Drug toxic effects disutility** | | | |
| Durvalumab plus tremelimumab | 0.206 (0.165 to 0.247) | Beta | 25, 26 |
| Durvalumab | 0.15 (0.12 to 0.18) | Beta | 25, 26 |
| EXTREME | 0.599 (0.48 to 0.719) | Beta | 25, 26 |
| **Other inputs** | | | |
| CT scans of the head and neck (per time) | 84.23 (41.99 to 160.29) | Gamma | 21 |
| Body surface area, m² | 1.86 (1.40 to 2.23) | Gamma | 2 |
| Body weight, kg | 70 (50–91) | Gamma | 2 |

Abbreviations: PFS, progression-free survival; OS, overall survival; HR, hazard ratio; PD, progressed disease; AEs, adverse events.

[a]Only expected values are presented for these survival model parameters.

[b]Overall total cost per patient regardless of treatment duration.

[c]The mean cost and utility toll of adverse events weighted by the frequency of occurrence.

net health benefit (INHB). A threshold of $150,000 for willingness to pay (WTP) was considered for QALYs [16]. According to recommendations, an ICER lower than the WTP threshold is considered cost-effective [28]. A discount rate of 3% was used to assess the cost and effectiveness [29].

### Sensitivity analysis

An analysis of one-way sensitivity for the input parameters was performed to verify the robustness of the model and to identify variables that significantly affected the analysis results. One-way sensitivity analyses were conducted by adjusting the input parameters individually to their minimum and maximum values within a 95% confidence interval (CI) of the referenced literature or by adjusting the base-case value by 20% to identify the factors that significantly influence economic outcomes.

For the probabilistic sensitivity analysis, 10,000 Monte Carlo simulations were conducted by sampling all input parameters simultaneously from predefined distributions. Gamma-distribution sampling was used for all costs. Beta distribution was used to sample the utility values and probabilities. To illustrate the probability of cost-effectiveness at various WTP thresholds, cost-effectiveness acceptability curves (CEACs) based on 10,000 iterations were plotted.

Statistical analyses were performed in R using the hesim and heemod packages, version 4.0.5.

## Results

### Base-case results

**Scenario 1 Regarding overall patients with R/M HNSCC.** Compared to Durva-Treme, the EXTREME dominant. Compared with Durva-mono, EXTREME increased the effectiveness by 0.012 QALYs and 0.037 overall life-years. With an additional cost of $85,420, the ICER was $7,310,878/QALY (Table 2). Compared to Durva-mono, Durva-Treme provided 0.010 QALYs and 0.017 overall life-years at a cost of $85,593, or an ICER of $8,931,776/QALY.

The INMB and INHB of EXTREME were $488 and 0.003 QALYs at a threshold of $150,000/QALY, respectively, compared with Durva-Treme. The INMB and INHB of EXTREME were -$83,668 and -0.558 QALYs compared to Durva-mono. Comparing Durva-Treme with Durva-mono, Durva-Treme yielded -0.561 QALYs and -$84,156 INMBs at a $150,000/QALY WTP threshold (Table 2).

**Scenario 2 Regarding patients with high PD-L1 expression.** Durva-mono was dominant over Durva-Treme and EXTREME. Compared to EXTREME or Durva-Treme, the INMB and INHB of Durva-mono were $51,188 and 0.341 QALYs, or $61,914 and 0.413 QALYs, respectively. In addition, Durva-Treme resulted in an increased effectiveness of 0.026 QALYs and an increase in the overall life-years of 0.027 compared with EXTREME, at an additional cost of $14,646. A corresponding ICER of $560,406/QALY was calculated. The INHB was -0.072 QALYs, and the INMB was -$10,726 at a WTP threshold of $150,000/QALY (Table 3).

### Sensitivity analysis

With a WTP threshold of 150,000/QALY and the current price, the cost-effectiveness acceptability curve (CEAC) showed that Durva-mono treatment was 100% (or near 100%) cost-effective for the total population or for patients with high PD-L1 expression. Moreover, Durva-mono was more cost-effective in patients with high PD-L1 expression, as evidenced by the CEAC of Durva-mono not intersecting with those of Durva-Treme or EXTREME as the WTP value increased (Fig 2).

Based on the one-way sensitivity analysis, the outcomes were most influenced by the HRs for OS and PFS, drug costs (cetuximab, tremelimumab, and durvalumab), average body surface area (BSA), and utilities for PFS and PD. Only a moderate or marginal association was observed between the remaining parameters and the model (S2 Fig).

## Discussion

The present study represents the first comparison of the health and economic outcomes of Durva-mono and Durva-Treme treatment regimens vs. EXTREME treatment regimens in patients with R/M HNSCC. According to KESTREL [13], both

**Table 2. Summary of cost and outcome results in the base-case analysis of overall patients.**

| Factor | EXTREME | DT | D | Incremental change | | |
| --- | --- | --- | --- | --- | --- | --- |
| | | | | EXTREME vs. DT | EXTREME vs. D | DT vs. D |
| **Cost, $** | | | | | | |
| Drug | 159,351 | 166,998 | 84,571 | 82,427 | 7,647 | -74,780 |
| Nondrug[a] | 398,301 | 390,827 | 387,661 | 3,166 | -7,474 | -10,640 |
| Overall | 557,652 | 557,825 | 472,232 | -173 | 85,420 | 85,593 |
| **Life-years** | | | | | | |
| Progression-free | 0.597 | 0.550 | 0.474 | 0.047 | 0.123 | 0.076 |
| Overall | 1.709 | 1.688 | 1.671 | 0.021 | 0.037 | 0.017 |
| **QALYs** | 1.202 | 1.200 | 1.191 | 0.002 | 0.012 | 0.010 |
| **ICERs, $** | | | | | | |
| Per life-year | NA | NA | NA | Dominated | 2,281,160 | 5,158,393 |
| Per QALY | NA | NA | NA | Dominated | 7,310,878 | 8,931,776 |
| INHB, QALY, at threshold 150,000 | NA | NA | NA | 0.003 | -0.558 | -0.561 |
| INMB, $, at threshold 150,000 | NA | NA | NA | 488 | -83,668 | -84,156 |

Abbreviations: D, durvalumab; DT, durvalumab plus tremelimumab; EXTREME, cetuximab, 5-fluorouracil, and either carboplatin or cisplatin; ICER, incremental cost-effectiveness ratio; INHB, incremental net health benefit; INMB, incremental net monetary benefit; NA, not applicable; QALYs, quality-adjusted life years.

[a]Nondrug cost includes the costs of adverse event management, subsequent best supportive care per patient, and follow-up care covering physician monitors, drug administration, and terminal care.

Durva-mono (HR, 0.96; 95% CI, 0.69–1.32) and Durva-Treme (HR, 1.05; 95% CI, 0.80–1.39) were not preferred over EXTREME regarding OS in patients with high PD-L1 expression, as well as in overall populations; a prolonged duration of response (DOR) was observed with Durva-mono (49.3%) and Durva-Treme (48.1%) compared to EXTREME (9.8%). However, the median PFS was longer for the EXTREME regimen. Based on the KESTREL trial, we assessed the cost-effectiveness of oncology drugs for United States payers so that they could make informed decisions regarding oncology drugs and define appropriate oncology treatment pathways [1]. Our results showed that Durva-mono was a cost-saving strategy in the United States, followed by the EXTREME regimen. However, Durva-Treme may not be an cost-effective option for the treatment of R/M HNSCC.

According to the results of the one-way sensitivity analysis, the average body surface area (BSA) is a substantial model input variable because cetuximab and other chemotherapy drugs are administered accordingly. Owing to the requirement for fixed dosages of Durva in overweight patients, the Durva-mono scheme appears to be more favorable. A higher cetuximab dose is required for overweight patients under the EXTREME scheme, resulting in significant waste. Consequently, patients with obesity having greater BSA levels would have more significant differences in ICER because they require a larger dose of cetuximab. There is the potential to improve the economic outcomes of patients with obesity through alternative payment programs, such as payment per patient. According to a previous report, cetuximab is too expensive to cover [30]. Additionally, a substantial effect of cetuximab cost on the outcome was observed in our analysis.

Monoclonal antibody unit costs also significant influenced. The cost of Durva-mono drugs is still higher in the United States than in other developed countries despite its cost-effectiveness. To help reduce the relatively high prices paid by Americans for drugs, the United States government previously proposed indexing Medicare prices to prices in other developed countries [31]. The unit prices of Durva and cetuximab may possibly decrease once this pricing strategy is implemented, improving their cost-effectiveness.

Durva-mono's dominance may be related to its cost, especially considering the relatively expensive cetuximab used in EXTREME. A lower unit price of cetuximab may make the EXTREME scheme more cost-effective. Furthermore,

**Table 3. Summary of cost and outcome results in the base-case analysis of PD-L1 high expression patients.**

| Factor | D | DT | EXTREME | Incremental change | | |
| --- | --- | --- | --- | --- | --- | --- |
| | | | | D vs EXTREME | D vs DT | DT vs EXTREME |
| **Cost, $** | | | | | | |
| Drug | 95,615 | 173,047 | 159,856 | -64,242 | -77,432 | 13,191 |
| Nondrug[a] | 449,459 | 419,900 | 418,445 | 31,014 | 29,559 | 1,455 |
| Overall | 545,073 | 592,947 | 578,301 | -33,228 | -47,874 | 14,646 |
| **Life-years** | | | | | | |
| Progression-free | 0.529 | 0.565 | 0.574 | -0.045 | -0.036 | -0.009 |
| Overall | 1.963 | 1.830 | 1.802 | 0.160 | 0.133 | 0.027 |
| **QALYs** | 1.385 | 1.291 | 1.265 | 0.120 | 0.094 | 0.026 |
| **ICERs, $** | | | | | | |
| Per life-year | NA | NA | NA | Dominated | Dominated | 534,502 |
| Per QALY | NA | NA | NA | Dominated | Dominated | 560,406 |
| INHB, QALY, at threshold 150,000 | NA | NA | NA | 0.341 | 0.413 | -0.072 |
| INMB, $, at threshold 150,000 | NA | NA | NA | 51,188 | 61,914 | -10,726 |

Abbreviations: D, durvalumab; DT, durvalumab plus tremelimumab; EXTREME, cetuximab, 5-fluorouracil, and either carboplatin or cisplatin; ICER, incremental cost-effectiveness ratio; INHB, incremental net health benefit; INMB, incremental net monetary benefit; NA, not applicable; QALYs, quality-adjusted life years.

[a]Nondrug cost includes the costs of adverse event management, subsequent best supportive care per patient, and follow-up care covering physician monitors, drug administration, and terminal care.

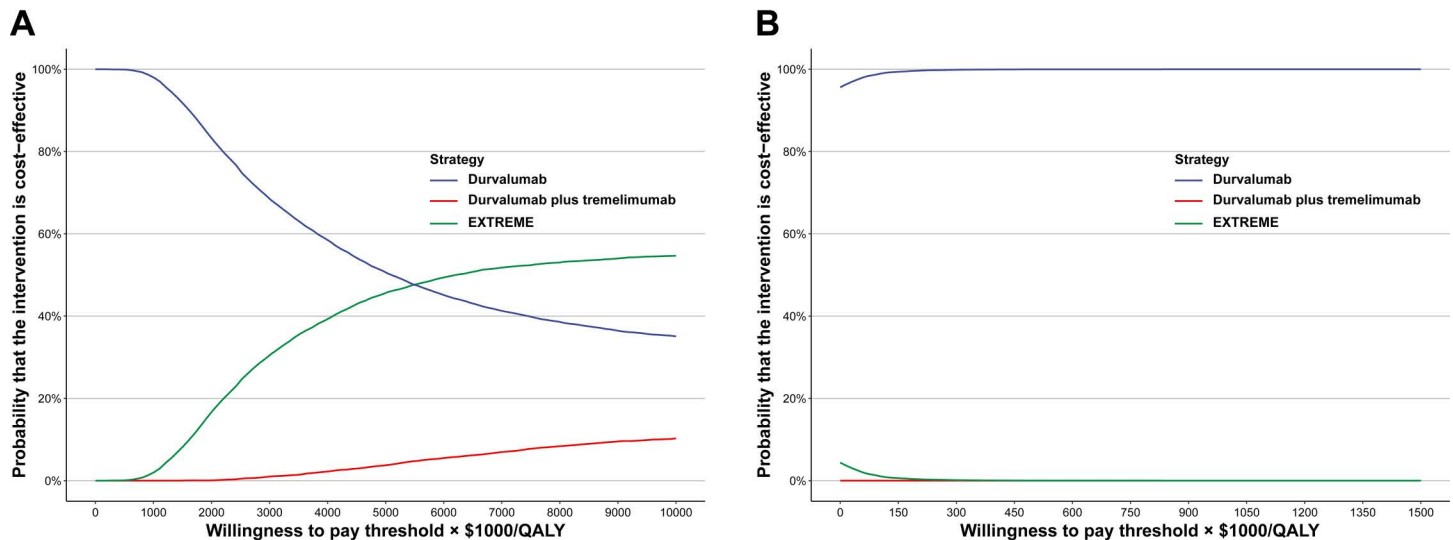

**Fig 2. Acceptability curves of cost-effectiveness for durvalumab, durvalumab plus tremelimumab and EXTREME.** (A) All randomized patients, (B) Patients with PD-L1 high expression. Abbreviations: QALY, quality-adjusted life-year; EXTREME, cetuximab plus platinum plus 5-fluorouracil.

Durva-Treme is not cost-effective because it does not provide adequate treatment results, and tremelimumab is expensive per unit. Additionally, no significant difference in efficacy was observed between the Durva-Treme and EXTREME regimens, especially in the subgroups with high PD-L1 levels. Therefore, the ICER increased sharply to $560,406/QALY. In-depth deliberations on these issues must be performed. The cost and benefit of treatment should be carefully

weighed when making a decision about treatment. A previously published study that evaluated first-line tremelimumab plus durvalumab and chemotherapy versus chemotherapy alone for metastatic non-small cell lung cancer showed similar results that tremelimumab plus durvalumab and chemotherapy was estimated to be less cost-effective than chemotherapy for patients with non-small cell lung cancer at a WTP threshold of $100,000 to $150,000/QALY in the United States [32].

Durvalumab and tremelimumab are immune checkpoint inhibitors that work together to enhance antitumor immunity [33]. Durvalumab blocks PD-L1, a protein that suppresses T-cell activity, thereby restoring the ability of the immune system to recognize and attack tumor cells [34]. Tremelimumab inhibits CTLA-4, a receptor that downregulates T-cell activation, leading to increased T cell proliferation and a stronger anti-tumor response [34]. Together, they create a synergistic effect, improving immune cell activation and altering the tumor microenvironment to favor immune-mediated tumor destruction [33,34]. They enhance T cell infiltration into tumors, reduce immunosuppressive factors, such as regulatory T cells and myeloid-derived suppressor cells, and modulate pathways involved in angiogenesis and tumor aggressiveness [33,34].

Healthcare insurers can use our study to make decisions for patients with R/M HNSCC in need of effective treatment based on a valid and informative economic assessment of the three treatment regimens. In this study, a three-state partitioned survival model structure was used, which is widely applied in the technological appraisals of oncology, including HNSCC [2,16,35]. Patient-level survival data from the KESTREL clinical trial were used to model progression-free and progressive disease state occupancy, ensuring high internal validity. Moreover, the various analyses explored within the parametric and structural assumptions of the economic analysis provides an accurate depiction of the robustness of the study conclusions by providing a clear and comprehensive evaluation of the assumptions that potentially influence the model results.

Our study has certain limitations. In our calculations, only grade ≥3 AEs were included. We found that the current AEs cost variation had a minimal effect on our model based on one-way sensitivity analyses. Because the management costs for grade 1–2 AEs were much lower than those for grade ≥3 AEs, we believe dismissing the grade 1–2 AEs fees is reasonable. In addition, because clinical trials are conducted in a selected population meeting inclusion and exclusion criteria, the resources used in cost-effectiveness analyses may not reflect resources in actual clinical practice. As KESTREL did not offer second-line or subsequent treatment information, we could not evaluate this in this study. Finally, we derived utilities based on previous studies because KESTREL did not offer results, which may have biased our findings. One-way sensitivity analyses were conducted to assess the uncertainty of utilities, and the results indicated that the proposed model was robust. Despite these limitations, our analysis was still reasonable.

## Conclusion

From the perspective of the United States payers, first-line treatment with Durva-mono is a cost-effective strategy for the overall population with R/M HNSCC, particularly in patients with high PD-L1 expression. A substantial decrease in the price of cetuximab may ultimately result in a favorable economic outcome for the EXTREME regimen. Nevertheless, Durva-Treme is unlikely to be a cost-effective first-line treatment option for R/M HNSCC and is associated with less clinical benefit and substantial cost. These findings may help doctors make better decisions regarding the treatment of R/M HNSCC.

## Supporting information

**S1 Fig. Model Fitting Analysis.**
(DOCX)

**S2 Fig. Tornado Diagram of One-Way Sensitivity Analyses.**
(DOCX)

**S1 Table.  Akaike Information Criterion and Bayesian Information Criterion Values from Each Survival Model.**
(DOCX)

**S2 Table.  Associated Costs and Disutility of Treatment-Related Adverse Events.**
(DOCX)

## Author contributions

**Conceptualization:** Huijuan Li, Xueyan Liang, Liju Huang, Yan Li.

**Data curation:** Huijuan Li, Xueyan Liang, Shiran Qin, Yan Li.

**Formal analysis:** Huijuan Li, Xueyan Liang, Shiran Qin, Liju Huang, Yan Li.

**Funding acquisition:** Yan Li.

**Investigation:** Huijuan Li, Xueyan Liang, Xiaoyu Chen, Liju Huang.

**Methodology:** Huijuan Li, Xueyan Liang, Yan Li.

**Project administration:** Huijuan Li, Xueyan Liang, Liju Huang, Yan Li.

**Resources:** Huijuan Li, Xiaoyu Chen, Liju Huang.

**Software:** Yan Li.

**Supervision:** Huijuan Li, Xueyan Liang, Xiaoyu Chen, Liju Huang.

**Validation:** Huijuan Li, Xueyan Liang, Xiaoyu Chen, Yan Li.

**Visualization:** Huijuan Li, Xueyan Liang, Xiaoyu Chen, Liju Huang.

**Writing – original draft:** Huijuan Li, Xueyan Liang.

**Writing – review & editing:** Liju Huang, Yan Li.

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
