## [Decision Letter · Decision Letter 0]

11 Mar 2025

PONE-D-24-42321First-line durvalumab alone or in combination with tremelimumab in metastatic head and neck squamous cell carcinoma: A cost-effectiveness analysisPLOS ONE

Dear Dr. Li,

Thank you for submitting your manuscript to PLOS ONE. After careful consideration, we feel that it has merit but does not fully meet PLOS ONE’s publication criteria as it currently stands. Therefore, we invite you to submit a revised version of the manuscript that addresses the points raised during the review process.

We look forward to receiving your revised manuscript.

Kind regards,

Keun-Yeong Jeong

Academic Editor

PLOS ONE

Journal Requirements:

https://journals.lww.com/md-journal/fulltext/2024/04190/cost_effectiveness_analysis_of_durvalumab,.67.aspx

In your revision ensure you cite all your sources (including your own works), and quote or rephrase any duplicated text outside the methods section. Further consideration is dependent on these concerns being addressed.

4. We note that your Data Availability Statement is currently as follows: “All relevant data are within the manuscript and its Supporting Information files.”

Please confirm at this time whether or not your submission contains all raw data required to replicate the results of your study. Authors must share the “minimal data set” for their submission. PLOS defines the minimal data set to consist of the data required to replicate all study findings reported in the article, as well as related metadata and methods (https://journals.plos.org/plosone/s/data-availability#loc-minimal-data-set-definition ).

If your submission does not contain these data, please either upload them as Supporting Information files or deposit them to a stable, public repository and provide us with the relevant URLs, DOIs, or accession numbers. For a list of recommended repositories, please see https://journals.plos.org/plosone/s/recommended-repositories .

Reviewers' comments:

Reviewer's Responses to Questions

**Comments to the Author**

1. Is the manuscript technically sound, and do the data support the conclusions?

Reviewer #1: Yes

Reviewer #2: Yes

Reviewer #3: Partly

2. Has the statistical analysis been performed appropriately and rigorously? 

Reviewer #1: I Don't Know

Reviewer #2: Yes

Reviewer #3: Yes

3. Have the authors made all data underlying the findings in their manuscript fully available?

Reviewer #1: Yes

Reviewer #2: Yes

Reviewer #3: Yes

4. Is the manuscript presented in an intelligible fashion and written in standard English?

Reviewer #1: Yes

Reviewer #2: Yes

Reviewer #3: Yes

5. Review Comments to the Author

Reviewer #1: In this manuscript by Li et al., the authors present a cost-effectiveness analysis comparing durvalumab monotherapy (Durva-mono), durvalumab + tremelimumab therapy (Durva-Treme), and the EXTREME regimen for recurrent or metastatic HNSCC. While the study addresses an important topic, I believe some aspects require further clarification before the manuscript can be accepted. In particular, understanding the conclusions from a biological perspective, along with HNSCC biology and the mechanisms of action of the therapies, could improve the interpretation and applicability of the analysis. Please see my comments below.

1. The manuscript concludes that Durva-mono is the more cost-effective strategy for overall patients with R/M HNSCC and for those with high PD-L1 expression but does not provide any biological insight into why a prolonged duration of response was observed for Durva-mono and Durva-Treme compared to the EXTREME regimen.

2. The manuscript does not consider factors such as post-progression therapies, patient genetics, immune landscape, etc., which may also influence the observed cost-effectiveness.

3. Although the authors define “PD-L1 High,” they do not mention or provide any information on how PD-L1 was measured in the patients. Several PD-L1 assays exist, and their results are not always interchangeable. Additionally, PD-L1 expression is heterogeneous among tumor cell subpopulations, so an overall estimation may fail to accurately capture the PD-L1 status of the tumor. The manuscript should specify which assay was used in KESTREL in the methods section and acknowledge potential limitations.

4. This manuscript relies solely on PD-L1 expression for its conclusions, which may overlook patients with low or negative PD-L1 expression who can still benefit from immunotherapy. Conversely, some patients with high PD-L1 expression do not respond to treatment. Other factors, such as the tumor microenvironment, immune system dynamics, and tumor mutational burden, should also be considered.

5. I believe the manuscript would benefit from a more detailed discussion of the mechanisms of durvalumab and tremelimumab, including their effects on immune cell activation, immune checkpoints, and the tumor microenvironment.

Minor comments:

1. There are several odd sounding sentences in the manuscript, a thorough proofreading is necessary before resubmission.

Reviewer #2: This study (First-line durvalumab alone or in combination with tremelimumab in metastatic head and neck squamous cell carcinoma: A cost-effectiveness analysis) is well-structured, methodologically sound, and highly relevant to oncology cost-effectiveness research. The use of partitioned survival models and comprehensive cost-effectiveness metrics enhances its robustness. Sensitivity analyses further strengthen the reliability of findings. The clinical relevance of evaluating Durva-mono and Durva-Treme in R/M HNSCC is significant. The manuscript effectively highlights key economic considerations for treatment decision-making. No revisions are necessary as the study is clear, impactful, and well-executed.

Reviewer #3: This paper demonstrates what combination of medicines are cost-effective as well as overall survival. Since the cost of treatments is crucial for patients, the findings are beneficial. The manuscript is well written, and conclusion is clear.

Here is one question:

Liu, Ho, Chen published similar report in 2023 (PMID: 37547328). Authors did not cite, but it would be nice to describe that paper and empathize the findings.

6. PLOS authors have the option to publish the peer review history of their article (what does this mean? ). If published, this will include your full peer review and any attached files.

**Do you want your identity to be public for this peer review?** For information about this choice, including consent withdrawal, please see our Privacy Policy .

Reviewer #1: No

Reviewer #2: **Yes: ** Dr. Nisat Alam

Reviewer #3: No

---

## [Author Response · Author response to Decision Letter 0]

5 Apr 2025

Dear Reviewers and Editors,

Thank you for your extremely efficient and earnest work on our manuscript. The comments are all valuable and very helpful for revising and improving our paper. The main corrections in the manuscript and the response to the Referees' comments are listed below point by point:

Reviewer #1: In this manuscript by Li et al., the authors present a cost-effectiveness analysis comparing durvalumab monotherapy (Durva-mono), durvalumab + tremelimumab therapy (Durva-Treme), and the EXTREME regimen for recurrent or metastatic HNSCC. While the study addresses an important topic, I believe some aspects require further clarification before the manuscript can be accepted. In particular, understanding the conclusions from a biological perspective, along with HNSCC biology and the mechanisms of action of the therapies, could improve the interpretation and applicability of the analysis. Please see my comments below.

1. The manuscript concludes that Durva-mono is the more cost-effective strategy for overall patients with R/M HNSCC and for those with high PD-L1 expression but does not provide any biological insight into why a prolonged duration of response was observed for Durva-mono and Durva-Treme compared to the EXTREME regimen.

Reply: Thank you for your suggestion. In this study, the KESTREL study evaluated durvalumab with or without tremelimumab, as well as the EXTREME regimen (platinum, 5-fluorouracil, and cetuximab) in patients with recurrent or metastatic head and neck squamous cell carcinoma (R/M HNSCC). The aim of this study is to determine whether durvalumab with or without tremelimumab or EXTREME were cost-effective for treating R/M HNSCC in the United States, either in the total population or in subgroups with PD-L1-high (TC ≥50% or IC ≥25%) expression, based on the survival outcome and adverse events results of KESTREL.

From the perspective of cost-effectiveness analysis, the manuscript’s conclusion can be critiqued without delving into biological explanations:

First, data on overall survival, progression-free survival, and adverse events were obtained from KESTREL. In this study, PSM was used, three health states were modeled, including progression-free survival (PFS), progressed disease (PD) and death. PFS state was the initial state of health for all patients, and during each cycle they could maintain their assigned health state or redistribute to another health state. At each time point, the area under the curve of the PFS was used to estimate the proportion of patients in the PFS state, and 1 minus the OS curve was used to estimate the proportion of patients in the death state. There was a PD state measured by the AUC between the PFS and OS curves. All of those data can be obtained from KESTREL, without any biological insight.

Second, in the conventional approach to QALYs the quality-adjustment weight for each health state is multiplied by the time in the state and then summed to calculate the number of QALYs. The advantage of the QALY as a measure of health output is that it can simultaneously capture gains from reduced morbidity (quality gains) and reduced mortality (quantity gains), and integrate these into a single measure. An anchored utility value was assigned to each health state in this partitioned survival model, with 0 being death and 1 being perfect health. In absence of data from the KESTREL trial, we derived utility values from published sources. A PD health utility value of 0.749 and a PFS health utility value of 0.805 were associated with HNSCC. Furthermore, grade ≥3 AEs disutility values were considered in accordance with relevant literature.

Third, drugs, routine follow-up, laboratory tests, best supportive care, management of adverse events, and end-of-life terminal care were considered in the analysis. The unit price for tremelimumab was calculated using Lexicomp Online. The Center for Medicare and Medicaid Services (CMS) provides other drug unit prices. The cost of terminal care per patient is $11126. Based on Tom's Inflation Calculator, all costs were inflated to 2022 US dollars using Medical-Care Inflation data.

Finally, the general rule when assessing Durva-Treme, Durva-mono and EXTREME is that the difference in costs is compared with the difference in consequences, in an incremental analysis. Cost-effectiveness analysis is of most use in situations where a decision-maker, operating with a given budget, is considering a limited range of options within a given field.

Furthermore, there is a range of important caveats in how subgroups are identified and presented, including the need for their biological plausibility, and many would argue that this can only be assured if the subgroup effects to be estimated are defined before the data are available. Based on the results of , only subgroups with PD-L1-high (TC ≥50% or IC ≥25%) expression were evaluated, and those cost-effectiveness analysis results were evaluated.

Hence, all of the evaluated data can be obtained and the full analysis were performed without other biological insight.

2. The manuscript does not consider factors such as post-progression therapies, patient genetics, immune landscape, etc., which may also influence the observed cost-effectiveness.

Reply: Thank you for your suggestion.

First, due to the KESTREL did not offer results second-line or subsequent treatments information, we cannot provide that information. Considering your suggestion, we have discussed in the limitation.

Second, in this study, the clinical data were obtained from KESTREL, a randomized, open-label, phase III study. Based on KESTREL, the randomization was stratified according to tumor cell (TC) PD-L1 expression (≥25% versus <25%,20,21), tumor location [oropharyngeal (OPC) or non-OPC], and smoking history (>10 or ≤10 pack-years). Patients with OPC were further stratified by HPV status (positive or negative). Based on that information, we considered the patient genetics, immune landscape are equal in both treatment arm.

3. Although the authors define “PD-L1 High,” they do not mention or provide any information on how PD-L1 was measured in the patients. Several PD-L1 assays exist, and their results are not always interchangeable. Additionally, PD-L1 expression is heterogeneous among tumor cell subpopulations, so an overall estimation may fail to accurately capture the PD-L1 status of the tumor. The manuscript should specify which assay was used in KESTREL in the methods section and acknowledge potential limitations.

Reply: Thank you for your suggestion. Considering your suggestion, we have provided the information of PD-L1 measured method and discussed the potential limitations of VENTANA PD-L1 (SP263) Assay in the section of limitation in the section of Patients and intervention.

PD-L1 expression was evaluated by VENTANA PD-L1 (SP263) Assay based immunohistochemistry. PD-L1 high (TC ≥50% or IC ≥25%) was defined as either ≥50% of TCs or ≥25% of ICs staining for PD-L1 at any intensity if >1% of the tumor area contained ICs or ≥50% of TCs or 100% of ICs staining for PD-L1 at any intensity if 1% of the tumor area contained ICs. In the definition of PD-L1 low, it is a condition that is not met by any of the characteristics of PD-L1 high. VENTANA PD-L1 (SP263) Assay is not approved with any other detection or instruments, and only Rabbit Monoclonal Negative Control Ig can be used.

4. This manuscript relies solely on PD-L1 expression for its conclusions, which may overlook patients with low or negative PD-L1 expression who can still benefit from immunotherapy. Conversely, some patients with high PD-L1 expression do not respond to treatment. Other factors, such as the tumor microenvironment, immune system dynamics, and tumor mutational burden, should also be considered.

Reply: Thank you for your suggestion. PFS and OS data for this study were obtained from KESTREL. We were unable to perform the related analysis due to the lack of detail provided by KESTREL on the PFS and OS of PD-L1 low. There was also no detailed PFS and OS information provided about other factors, such as the tumor microenvironment, immune system dynamics, or tumor mutational burden.

5. I believe the manuscript would benefit from a more detailed discussion of the mechanisms of durvalumab and tremelimumab, including their effects on immune cell activation, immune checkpoints, and the tumor microenvironment.

Reply: Thank you for your suggestion. We have discussed the mechanisms of durvalumab and tremelimumab in the section of discussion.

Durvalumab and tremelimumab are immune checkpoint inhibitors that work together to enhanced antitumor immunity [33]. Durvalumab blocks PD-L1, a protein that suppresses T cell activity, thereby restoring the ability of immune system to recognize and attack tumor cells [34]. Tremelimumab inhibits CTLA-4, a receptor that downregulates T cell activation, leading to increased T cell proliferation and a stronger anti-tumor response [34]. Together, they create a synergistic effect, improving immune cell activation and altering the tumor microenvironment to favor immune-mediated tumor destruction [33, 34]. They enhance T cell infiltration into tumors, reduce immunosuppressive factors like regulatory T cells and myeloid-derived suppressor cells, and modulate pathways involved in angiogenesis and tumor aggressiveness [33, 34].

Minor comments:

1. There are several odd sounding sentences in the manuscript, a thorough proofreading is necessary before resubmission.

Reply: Thank you for your suggestion. The overall language expression of this study has been improved by the Editage.

Reviewer #2: This study (First-line durvalumab alone or in combination with tremelimumab in metastatic head and neck squamous cell carcinoma: A cost-effectiveness analysis) is well-structured, methodologically sound, and highly relevant to oncology cost-effectiveness research. The use of partitioned survival models and comprehensive cost-effectiveness metrics enhances its robustness. Sensitivity analyses further strengthen the reliability of findings. The clinical relevance of evaluating Durva-mono and Durva-Treme in R/M HNSCC is significant. The manuscript effectively highlights key economic considerations for treatment decision-making. No revisions are necessary as the study is clear, impactful, and well-executed.

Reply: I am deeply grateful for the time and effort you have dedicated to reviewing my article. No revisions are necessary as the study is clear, impactful, and well-executed has filled me with immense joy, and I owe this success to your thoughtful evaluation and encouraging feedback.

Reviewer #3: This paper demonstrates what combination of medicines are cost-effective as well as overall survival. Since the cost of treatments is crucial for patients, the findings are beneficial. The manuscript is well written, and conclusion is clear.

Here is one question:

Liu, Ho, Chen published similar report in 2023 (PMID: 37547328). Authors did not cite, but it would be nice to describe that paper and empathize the findings.

Reply: Thank you for your suggestion. PMID: 37547328 evaluated first-line tremelimumab plus durvalumab and chemotherapy versus chemotherapy alone for metastatic non-small cell lung cancer, and this study showed that chemotherapy was not cost-effective compared with chemotherapy at a WTP threshold of $ 100,000 to $ 150,000 per QALY in the United States. Our study also showed similar that EXTREME was not cost-effective in R/M HNSCC. Considering your suggestion, we have discussed in the section of discussion.

Previously published study evaluated first-line tremelimumab plus durvalumab and chemotherapy versus chemotherapy alone for metastatic non-small cell lung cancer showed similar results that tremelimumab plus durvalumab and chemotherapy was estimated to be less cost-effective than chemotherapy for patients with non-small cell lung cancer at a WTP threshold of $ 100,000 to $ 150,000 per QALY in the United States [32].

---

## [Decision Letter · Decision Letter 1]

21 Apr 2025

First-line durvalumab therapy alone or in combination with tremelimumab for metastatic head and neck squamous cell carcinoma: A cost-effectiveness analysis

PONE-D-24-42321R1

Dear Dr. Li,

We’re pleased to inform you that your manuscript has been judged scientifically suitable for publication and will be formally accepted for publication once it meets all outstanding technical requirements.

Kind regards,

Keun-Yeong Jeong

Academic Editor

PLOS ONE

Reviewers' comments:

Reviewer's Responses to Questions

**Comments to the Author**

1. If the authors have adequately addressed your comments raised in a previous round of review and you feel that this manuscript is now acceptable for publication, you may indicate that here to bypass the “Comments to the Author” section, enter your conflict of interest statement in the “Confidential to Editor” section, and submit your "Accept" recommendation.

Reviewer #1: All comments have been addressed

Reviewer #3: All comments have been addressed

2. Is the manuscript technically sound, and do the data support the conclusions?

Reviewer #1: Yes

Reviewer #3: Yes

3. Has the statistical analysis been performed appropriately and rigorously? 

Reviewer #1: I Don't Know

Reviewer #3: Yes

4. Have the authors made all data underlying the findings in their manuscript fully available?

Reviewer #1: Yes

Reviewer #3: Yes

5. Is the manuscript presented in an intelligible fashion and written in standard English?

Reviewer #1: Yes

Reviewer #3: Yes

6. Review Comments to the Author

Reviewer #1: I am satisfied with the revised paper and recommend it for acceptance. While I would have appreciated some biological insight into why a prolonged duration of response was observed with Durva-mono and Durva-Treme compared to the EXTREME regimen as this could have offered a possible explanation, I understand that the primary aim of this study is to assess cost-effectiveness. Therefore, I am happy to recommend it for acceptance without additional explanation.

Congratulations!

Reviewer #3: (No Response)

7. PLOS authors have the option to publish the peer review history of their article (what does this mean? ). If published, this will include your full peer review and any attached files.

**Do you want your identity to be public for this peer review?** For information about this choice, including consent withdrawal, please see our Privacy Policy .

Reviewer #1: No

Reviewer #3: No

---

## [Editor Report · Acceptance letter]

PONE-D-24-42321R1

PLOS ONE

Dear Dr. Li,

I'm pleased to inform you that your manuscript has been deemed suitable for publication in PLOS ONE. Congratulations! Your manuscript is now being handed over to our production team.

Kind regards,

on behalf of

Dr. Keun-Yeong Jeong

Academic Editor

PLOS ONE